# Characterization and Expression of Holothurian Wnt Signaling Genes during Adult Intestinal Organogenesis

**DOI:** 10.3390/genes14020309

**Published:** 2023-01-25

**Authors:** Noah A. Auger, Joshua G. Medina-Feliciano, David J. Quispe-Parra, Stephanie Colón-Marrero, Humberto Ortiz-Zuazaga, José E. García-Arrarás

**Affiliations:** 1Department of Biology, University of Puerto Rico, Rio Piedras Campus, San Juan 00925, Puerto Rico; 2Department of Computer Science, University of Puerto Rico, Rio Piedras Campus, San Juan 00925, Puerto Rico

**Keywords:** Wnt genes, regeneration, organogenesis, echinoderm, sea cucumber

## Abstract

Wnt signaling has been shown to play multiple roles in regenerative processes, one of the most widely studied of which is the regeneration of the intestinal luminal epithelia. Most studies in this area have focused on self-renewal of the luminal stem cells; however, Wnt signaling may also have more dynamic functions, such as facilitating intestinal organogenesis. To explore this possibility, we employed the sea cucumber *Holothuria glaberrima* that can regenerate a full intestine over the course of 21 days after evisceration. We collected RNA-seq data from various intestinal tissues and regeneration stages and used these data to define the *Wnt* genes present in *H. glaberrima* and the differential gene expression (DGE) patterns during the regenerative process. Twelve *Wnt* genes were found, and their presence was confirmed in the draft genome of *H. glaberrima.* The expressions of additional Wnt-associated genes, such as *Frizzled* and *Disheveled*, as well as genes from the Wnt/β-catenin and Wnt/Planar Cell Polarity (PCP) pathways, were also analyzed. DGE showed unique distributions of Wnt in early- and late-stage intestinal regenerates, consistent with the Wnt/β-catenin pathway being upregulated during early-stages and the Wnt/PCP pathway being upregulated during late-stages. Our results demonstrate the diversity of Wnt signaling during intestinal regeneration, highlighting possible roles in adult organogenesis.

## 1. Introduction

The Wnt gene family encodes ligands that bind to the Frizzled (Fzd) family of cell surface receptors. This, in turn, activates canonical and noncanonical Wnt signaling pathways that are known to modulate various cellular processes such as morphogenesis, cell fate specification and proliferation during embryonic/adult development, and regeneration [1,2,3]. The canonical pathway, known as Wnt/β-catenin, can regulate proliferation, differentiation, and survival by affecting gene regulation through translocation of β-catenin into the nucleus that then interacts with the T cell factor/lymphoid enhancer factor (TCF/LEF) family [4,5,6,7].

The noncanonical pathway, known as the Planar Cell Polarity (PCP) pathway, can regulate cell migration and polarity by cytoskeleton rearrangements and activation of the transcription factors c-JUN and AFT2 [8,9]. These Wnt activated pathways share genes such as *Wnt* and *Fzd*, but some genes are unique to a pathway. For example, the Wnt/PCP pathway contains the so called “core” proteins, i.e., Celsr, Vangl, Prickle and Ankrd6 [10,11]. For a more in-depth view of the genes involved in and the signaling dynamics of the Wnt/β-catenin and Wnt/PCP pathways, see Appendix A. 

The cellular processes described above that are modulated by Wnt signaling can be species, organ, and cell specific. They also vary depending on developmental stage or whether the regenerative process is for cell maintenance (homeostatic) or injury response. One of the most studied functions of Wnt signaling is the self-renewal of the intestine luminal epithelium. Studies on distant groups, such as insects (Drosophila) and mammals, even hint at common mechanisms involving Wnt mediating the control of luminal epithelium regeneration [12,13,14]. In mammals, several *Wnt*s have functions during gastrointestinal stem cell homeostasis, such as *Wnt1* [15], *Wnt2b* [16], *Wnt3a* [17], *Wnt5a* [18], *Wnt6* [19], and *Wnt9b* [20]. In Drosophila, Wnt might play a role in stem cell maintenance [21], but more recent studies suggest a critical role for enterocyte homeostasis [22,23,24].

Although the above studies characterize Wnt signaling during the maintenance (homeostatic regeneration) of certain cell types in the intestine, Wnt signaling is more complex and can be involved in the regeneration of a complete adult organ. Therefore, to obtain a holistic view of Wnt signaling during regenerative organogenesis, we employed a well-suited model organism, the sea cucumber *Holothuria glaberrima* [25]. This echinoderm, like most members of the Echinodermata phylum, has remarkable regenerative properties. The histological and cellular events during sea cucumber intestinal regeneration are well-characterized and have been described in previous publications [25,26,27,28]. Below, we provide a brief overview of the stages of intestinal organogenesis in *H. glaberrima* (Figure 1). To initiate regenerative organogenesis in the lab, we induced autotomy (evisceration) by injecting sea cucumbers with KCl. Subsequently, the sea cucumber severed its connections to the intestine at the esophagus and cloaca, expelling the intestine through the cloaca, while leaving most of the mesentery attached at one end to the body wall. Regeneration immediately begins after evisceration in the form of wound healing along the torn mesenterial edge that serves as the cell source for the regenerating rudiment, composed of coelomic mesothelium and connective tissue. Between 3- and 7-days post-evisceration (dpe), a blastema-like structure forms. Around 14-dpe, the luminal epithelium derived from the esophagus (anterior) and cloaca (posterior) invade the rudiment and connect to form a continuous lumen at around 21-dpe.

The molecular processes that define some of the cellular events during sea cucumber intestinal organogenesis were recently investigated. *Wnt* genes are seen as key molecular regulators [29]. For example, microarray analysis and qPCR showed that *Wnt9* was upregulated at 3-, 7-, and 14-dpe during intestinal regeneration [30]. Moreover, in situ hybridization confirmed high expression of *Wnt9* in the luminal epithelium of early regenerates [31]. More recently, qPCR confirmed *Wnt6* upregulation during early intestinal regeneration [32]. It was even shown that Wnt/β-catenin signaling controls cell proliferation but not cell differentiation or apoptosis after RNAi of β-catenin during early-stage regeneration [33]. 

Wnt signaling is also upregulated during intestinal regeneration in two other holothurians: *Apostichopus japonicus* and *Eupentacta fraudatrix*. In *A. japonicus,* it was shown by qPCR that *WntA* and *Wnt6* were upregulated in early- and late-stage regeneration stages [34,35]. A different study, also using qPCR, showed that *Wnt7* and *Wnt8*, along with *Fzd7* and *Dishevelled* (*Dvl*), were upregulated at early-stage regeneration but either lowly expressed or downregulated in the later-stages [36,37]. Studies of intestinal regeneration in *E. fraudatrix* showed the *Wnt4*, *Wnt6*, *Wnt16*, *Fzd1/2/7*, *Fzd4*, and *Fzd5/8* genes to be differentially expressed by qPCR in early- and late-stage regeneration [38]. All these studies provide some insight into the presence and expression of *Wnt*s and the genes involved in Wnt pathways during intestinal regeneration. However, they also present problems when trying to make comparisons. To name a few, some studies focused only on some *Wnt* or *Fzd* genes while neglecting downstream genes in the signaling cascade. In other studies, the exact developmental stage of the tissue remained unclear and, in some cases, even the organ or tissue composition of the sample could not be definitively determined from the authors’ description. The other major problem is that these studies often used inadequate tissue comparisons as controls. All these problems create a patched landscape of studies of Wnt signaling during intestinal regeneration.

Therefore, the present work focuses on *Wnt* gene identification and the expression of Wnt signaling genes during intestinal organogenesis from a holistic point of view. To do this, we first identified the *Wnt* genes found in the sea cucumber genome and compared them to those of other echinoderm species. We next characterized their expression at various regeneration stages and in different tissue types. Lastly, we tried to determine which Wnt signaling pathways were active at different timepoints. Our study adds to the growing field of research of intestinal regeneration in echinoderms by supplying new spatiotemporal timepoints with appropriate controls for early- and late-stage regenerates, as well as tying together the available data on Wnt signaling during intestine regeneration in echinoderms and other species. We hope that this extensive spatiotemporal RNA-seq data will aid in the identification of master genetic regulators of organ regeneration that can later be assayed for downstream functional analysis.

## 2. Materials and Methods

### 2.1. Animals and Treatment

The animal handling and dissection methods applied in this study have been described previously [32]. In brief, adult sea cucumbers were collected from northern Puerto Rican shores at the coordinates 18°28′11.2″ N 66°07′07.9″ W and transported to the laboratory. They were placed in aerated sea water aquaria at room temperature until the time of evisceration, at which point they were given intracoelomic injections of 0.35 M KCl. The eviscerated sea cucumbers were then placed back into the aquaria and left to regenerate. They were later anesthetized by ice water immersion for 45 min and dissected under RNase-free conditions. Timepoints of 1- and 3- days of regeneration and mesenteries from normal, non-eviscerated animals were used in a previous study [32]. Here, we included additional regeneration stages for 12-hours post evisceration (hpe) and 3-, 7-, 14-, and 21-dpe. In addition, we included samples from normal large intestine. Regenerating intestines at 14-dpe were separated into three sections (Figure 1): the anterior portion, where the lumen had already formed, the middle rudiment (no lumen), and the posterior portion that had also formed a lumen. Intestines containing luminal epithelium were lifted from the body wall using forceps and cut from their mesenterial connections with surgical scissors. The same dissection method was used for non-eviscerated intestine to isolate it from the mesentery. Dissected tissues were placed in an Eppendorf tube with RNAlater solution and stored at 4 °C until the RNA extraction procedure was carried out. Dissected tissues were pooled together so that each sample contained tissues from two different specimens.

RNA extraction was done as published earlier [32]. A combination of the method established by Chomczynski (1993) using Tri-reagent (N.93289, Sigma, St. Louis, USA) and the RNAeasy mini kit (Qiagen, Hilden, Germany) was used for RNA extraction. Three different samples were processed for each stage, and each extraction consisted of pooled tissue samples from two organisms at the same stage. The concentration and quality of the extracted RNA were assessed using a 2100 Bioanalyzer (Agilent Technologies, Santa Clara, USA). Only samples that showed a concentration greater than 200 ng/μL and an RNA Integrity Number (RIN) value of ≥8 were used for sequencing. The obtained RNA was sequenced at the Sequencing and Genotyping Facility of the University of Puerto Rico. Libraries were constructed based on the TruSeq Stranded mRNA Library Prep Kit (Illumina, San Diego, USA), and paired-end sequencing was performed using an Illumina NextSeq 500 sequencer.

### 2.2. Wnt Characterization and Manual Annotation

Intestine transcriptomic data were first utilized for mapping all the potential *Wnt* sequences using amino acid sequences from *Strongylocentrotus purpuratus* and *Eupentacta fraudatrix*, obtained from EchinoBase and NCBI. Sequences from these species were utilized as queries against the sea cucumber peptide transcriptomic database, available at NCBI (BioProject: PRJNA660762) [32]. The ORF of the obtained sequences were further characterized by BLASTp against the NCBI RefSeq database. The resulting *Wnt* transcripts were then used to assess if further genes from the *Wnt* family were present in the genome. Additionally, these were manually annotated in the draft genome [39] of the sea cucumber, available at NCBI (ID: GCA_009936505.2), to confirm that all were distinct genes. For the manual annotation, the sequences obtained from the transcriptome were mapped against the draft genome using BLASTn and tBLASTn for nucleotide and amino acid sequences, respectively. We used both BLAST results to manually identify the exons of all the *Wnt* genes identified. Based on the coordinates obtained, we extracted the sequences for each exon of all the *Wnt* genes using the getFasta feature of BedTools v.2.30.0. Exons of each Wnt were then joined together using a custom Perl script described in a previous study [39].

### 2.3. Phylogenetic Analysis

The *Wnt* gene sequences obtained from the manual genome annotation were used to perform a phylogenetic analysis. Given that this annotation was done utilizing a draft genome assembly, we wanted to confirm that our characterization has been done properly and that each *Wnt* was grouped with those of other echinoderm species. This would also allow us to assess the conservation of the *Wnt* gene family with other echinoderms. Here, we followed protocols previously employed [39], starting by using MAFFT v.7.312 [40] to perform a multiple sequence alignment of Wnt amino acid sequences from *S. purpuratus, Patiria miniata, Lytechinus variegatus, Apostichopus japonicus,* and *E. fraudatrix.* All of these were obtained from NCBI or EchinoBase (Appendix A). Aligned sequences were utilized to create a phylogenetic tree by comparative analysis using IQTree v.2.0.3 [41] and RAXmL v.8.2.12 [42]. The complete methods and code for our phylogenetic analysis can be found in the Github repository of this project (https://github.com/devneurolab/HgWnt2023) (Accessed: 1 April 2023). 

### 2.4. Differential Gene Expression (DGE)

To assess DGEs of the candidate genes, we utilized previously available data from 1- and 3-dpe timepoints deposited at NCBI (ID: PRJNA660762) [32], along with additional sequenced samples for stages 12-hpe, 3-, 7-, 14, and 21-dpe and normal intestine, as mentioned before. For the differential expression analysis, 14-dpe samples were separated based on the three isolated regions: the anterior and posterior portions where the lumen had already formed and the middle section where no lumen was present. Moreover, distinct tissue samples were utilized as controls, depending on the stage. For instance, samples where no luminal epithelium was present were compared to the mesentery tissue of non-eviscerated sea cucumbers, whereas those with luminal epithelium were compared to intestinal tissue of non-eviscerated sea cucumber. Further details of these RNA-seq are being prepared for publication. 

DGE was performed based on previously reported protocols [32]. Reads were initially trimmed using Trimmomatic v.0.39 [43] (ILLUMINACLIP:{}:2:40:15 LEADING:20 TRAILING:20 SLIDINGWINDOW:4:15 MINLEN:35) and then quantified using Salmon v.0.8.2 [44] with default parameters and the dumpEq flag that produces equivalence classes. Since the focus of this study was not to analyze the complete expression profile of all the stages, but rather, to determine the expression of candidate genes, we utilized previously assembled transcriptome deposited at NCBI (ID: GIVL00000000.1). Equivalence classes generated with Salmon were then passed through Corset v.1.09 [45] with default parameters to hierarchically cluster contigs by sequence similarity and expression class. Then, DESeq2 [46] was used to perform differential expression analyses based on Corset clusters from the read counts generated. Samples with less than 30 read counts were filtered from the biological replicates of each sample. To identify the candidate genes on the transcriptome, we used the characterized *Wnt* genes sequences and annotated the transcriptome using UniProt database and predictions from *S. purpuratus* (NCBI ID: GCF_000002235.5). The complete script of the differential expression analysis can be found in the following GitHub: https://github.com/devneurolab/HgWnt2023 (accessed on 4 January 2023).

## 3. Results

### 3.1. Wnt Genes in H. glaberrima

#### 3.1.1. Wnt Gene Identification and Manual Annotation

Our initial identification of *H. glaberrima Wnt* genes was done using *Wnt* gene sequences from the sea urchin *S. purpuratus* and the sea cucumber *E. fraudatrix.* These sequences were used to probe our *H. glaberrima* transcriptome database for sequences that showed considerable similarities. The obtained sequences were further characterized by BLAST against the NCBI non-redundant database (Appendix A). Comparisons were also made between *Wnt* genes from three additional echinoderm species, i.e., the green sea urchin *Lytechinus variegatus,* the sea cucumber *A. japonicus,* and the sea star *Anchaster planci*, all of which demonstrated high levels of similarity with their corresponding homologs. Thus, we were able to identify the presence of transcripts for 12 different *Wnt* genes: *Wnt1*, *Wnt2*, *Wnt3*, *Wnt5*, *Wnt6*, *Wnt7*, *Wnt9*, *Wnt10*, *Wnt16*, *WntA,* and a possible duplication of *Wnt4* (*Wnt4a* and *Wnt4b*).

The identification of these transcripts allowed us to manually annotate all of them in our recently published draft genome of *H. glaberrima* [39] (Figure 2). Confirming the complete sequence of the identified transcripts, exons of annotated *Wnt* transcripts were located at distinct, non-overlapping regions in the genome, where we obtained their sequence from start to stop codon. Although most of the *Wnt*s exons spanned across distinct scaffolds, the order of the exons with exact matches for nucleotide and amino acid sequences was maintained for each *Wnt***,** providing sufficient information to map each exon to its respective *Wnt* gene. The annotations showed that all *Wnt* family genes in *H. glaberrima* ranged from four to six exons, with similar lengths compared to their intronic regions, which differed from gene to gene. To allow such comparisons, in Figure 2, we depict the length based on unit conversions; therefore, exon and intron sizes are based on the base pairs length.

A genomic analysis also allowed us to further confirm the duplication of *Wnt4* by determining the distinct distribution of exons in the genome (Figure 2) and by the alignment of sequences with an 89% percent degree of similarity (Appendix A). Thus, we concluded this to be a biological duplication, rather than a sequencing or assembly artifact.

It is important to highlight that (i) all *Wnt* genes that were found in the transcriptome database were also found in the draft genome, and (ii) that no new *Wnt* genes were found in the draft genome that were not present in our transcriptome database. This was true for *Wnt8* and *Wnt11*, which were not detected in the transcriptome or the genome, even after attempting to find a homologous sequence by using multiple sequences from other echinoderms.

#### 3.1.2. Comparison of Wnt Genes in Echinodermata

The protein coding sequences of the *Wnt* genes annotated in our *H. glaberrima* genome were compared to those found in NCBI or EchinoBase for species from other classes of the Echinodermata clade. Only sequences from three classes were reported: Asteroidea (sea stars), Echinoidea (sea urchins and sand dollars), and Holothuroidea (sea cucumbers). Therefore, *Wnt* sequences for the Crinoidea (sea lilies) and Ophiuroidea (brittle stars) classes were not included in our cross-species comparisons.

Our phylogenetic analysis was performed using *Wnt* sequences from sea cucumbers *A. japonicus* and *E. fraudatrix (Holothuroidea),* sea urchins *L. variegatus* and *S. purpuratus* (Echinoidea), and the sea star *Patiria miniata* (Asteroidea). The results showed that all our sequences were in the same clade of their proposed *Wnt* homologs, confirming the discovery and annotation of the *Wnt* genes mentioned above (Figure 3). Furthermore, it demonstrated a high conservation of these within the major classes of the Echinodermata clade, obtaining bootstrap values of more than 95 for each specific *Wnt* gene type (Figure 3, nodes with pink circles). Similarly, for those *Wnt*s that are present in two or more holothurian species, these sequences were grouped in sister clades to those of echinoid species.

As it is known that *Wnt* genes are conserved in clusters, we wanted to know if clusters persist in *H. glaberrima* (Figure 4). Initially, we searched for the genomic coordinates of the *Wnt* genes within the genomes of *L. variegatus* and *S. purpuratus* for the sake of comparison and found *Wnt9, Wnt3, Wnt1, Wnt6, and Wnt10* clustered together. Based on the fragmentation of the currently available *H. glaberrima* draft genome, we could only confirm the cluster conservation of *Wnt1, Wnt6,* and *Wnt10.* It appears that the cluster is not only conserved but also oriented in the same manner as that of *L. variegatus.* However, the cluster in *S. purpuratus* shows a different orientation for *Wnt1* and *Wnt6* compared to both *L. variegatus* and *H. glaberrima.* Similarly, when the same cluster is compared with that in *Drosophila,* the structure and order are maintained, albeit with a change in *Wnt6* orientation.

### 3.2. Expression of Wnt Signaling Genes 

#### 3.2.1. Wnt Expression

To determine the differential expression of *Wnt* genes during intestinal regeneration, we performed differential gene expression (DGE) analyses using various RNA-seq timepoints. These included previously published data from normal mesentery and from 1- and 3-dpe rudiments [32], as well as unpublished transcriptomes that include normal uneviscerated large intestine, 12 h post-evisceration (hpe), 3-, 7-, 14- and 21-dpe. At 14-dpe, the regenerating rudiment is in the process of forming the lumen. Therefore, tissues from 14-dpe animals were used to make three different transcriptomes, corresponding to 14-dpe anterior (with lumen), 14-dpe rudiment (middle, no lumen), and 14-dpe posterior (with lumen). Since the transcriptomes correspond to different batches, some timepoints were repeated to be able to assess and eliminate possible batch effects. With all these timepoints, we performed a principal component analysis (PCA) to determine the relationship of gene expression at different regeneration stages (Appendix A). The PCA results showed two distinct groups that represent the normal mesentery and normal intestine. The PCA results for regenerating samples displayed a major difference between early-stage regenerates (12-hpe, 1- and 3-dpe) and late-stage regenerates (14- and 21-dpe) with the 7-dpe samples separating the two groups.

The differential expression of *Wnt* genes during intestinal regeneration was analyzed using all RNA-seq timepoints mentioned (Figure 5). Therefore, two different controls were used for these analyses. Regenerating intestinal stages that lacked a lumen (and, therefore, a luminal epithelium) were compared to the normal mesentery, while regenerating intestines that had a lumen (and, therefore, a luminal epithelium) at 14-dpe anterior and posterior and 21-dpe were compared to normal intestine.

*Wnt3*, *Wnt6,* and *Wnt9* were found to be upregulated both in early- and late-stage regeneration. However, only advanced regenerative intestines that had luminal epithelium showed differential expression of *Wnt4a*, *Wnt4b*, *Wnt5,* and *WntA*. Only one Wnt, *Wnt7*, showed a decrease in expression, specifically, at early-stage regenerating intestine (12-hpe) when compared to the normal mesentery and in regenerating intestines at advanced stages when compared to normal intestine. An interesting finding was the difference in *Wnt* gene expression between 14-dpe anterior and 14-dpe posterior intestine. In the anterior intestine with luminal epithelium, *Wnt6* was uniquely upregulated, while in posterior tissue, *Wnt5* and *Wnt10* were uniquely upregulated (Figure 5).

#### 3.2.2. Expression of Wnt-Associated Genes

The availability of differential gene expression data from different regeneration stages and tissues offers the additional possibility of exploring the expression profile of other genes related to Wnt signaling pathways. Thus, we explored the expression of the members of the Wnt receptor family Frizzled (Fzd) and the Wnt signaling pathway protein Dishevelled (Dvl) (Figure 6). Our results showed that while *Dvl-3* was expressed in the intestinal transcriptomes, there was no clear differential expression among the different stages or tissues. In contrast, three out of the five *Fzd*s that were found in our intestinal transcriptomes showed differential expression. *Fzd1* and *Fzd10* were upregulated in the late regeneration stages, i.e., posteriorly *Fzd10* at 14-dpe while anteriorly *Fzd1* at 21-dpe. *Fzd4* was down-regulated during early-stage regeneration in the rudiment but was then up-regulated in late regenerating intestines when compared to the normal intestine.

#### 3.2.3. Wnt/β-Catenin and Wnt/PCP Pathway Gene Expression

Two major branches of Wnt signaling pathways exist, i.e., the canonical and the non-canonical Wnt signaling pathways, which can mediate different functions in regenerative processes [2]. A brief overview of both pathways can be found in Appendix A. The differential gene expression data were also used to explore the possibility that a particular pathway could be associated with an individual regeneration stage or tissue. For this, we searched for the transcripts of genes associated with either the canonical (Wnt/β-catenin) pathway or non-canonical (Wnt/PCP) pathway in our transcriptomic database and then analyzed these for differentially expressed genes. First, we assessed genes in the β-catenin pathway (Figure 7). In the early regenerative stages, two genes, *Groucho* (a repressor of TCF/LEF) and *Kremen1* (an inhibitor of the Wnt/Fzd/LRP6 complex), were down-regulated from 12-hpe to 3-dpe when compared to the normal mesentery. *DKK3* was upregulated only in the 12-hpe stage following evisceration, while *Myc* (a target gene) was upregulated during early-stage regeneration from 12-hpe to 3-dpe. Conversely, in the late regenerative stages, *Kremen1* and *Axin2* (a target gene and self-inhibitor) were upregulated when compared to the normal intestine. Additional genes associated with the canonical pathway that were found to be upregulated in some late-regenerative stages were *Twist* and *Slug* (specifically in the anterior 14-dpe stage) and *EDNRA*, *SP5*, and *BAMBI* in anterior and posterior 14-dpe, as well as in 21-dpe.

As for the Wnt/PCP pathway, several associated genes were downregulated during the early regenerative stages, including *Lamc1* and *Ror1* at 12-hpe and *c-Jun* and *Ctnnal1* from 12 hpe to 3-dpe (Figure 8). Two genes were over-expressed at the same time periods, i.e., *Mfhas1* at 12-hpe and *Rac1* from 12-hpe to 3-dpe. In the later stages of regeneration, we found several Wnt/PCP associated genes to be over-expressed, including the core proteins Ankrd6 and Vangl2 and other genes such as *Pvr*, *Mfhas1*, *Lamc1*, *Daam2*, and *Ror1*. Only one Wnt/PCP associated gene, *Map1ic3b*, was found to be down-regulated in the late regenerative stages (14-dpe anterior and posterior) when compared to the normal intestine.

## 4. Discussion

### 4.1. Comparison of Wnt Genes in Echinodermata

In the present study, we found 12 *Wnt* genes in the genome of *H. glaberrima*: *Wnt1, Wnt2, Wnt3, Wnt4a, Wnt4b, Wnt5, Wnt6, Wnt7, Wnt9, Wnt10, Wnt16, and WntA.* Surprisingly, all these genes were shown to be expressed in the intestinal tissue, where they were initially identified, and no other *Wnt* gene was found in the genome. This is somewhat unexpected, as *Wnt genes* are involved in diverse developmental and biological processes; thus, we expected to find *Wnt* genes in the genome that were not expressed in the intestinal transcriptomes.

Further, the manual annotation and characterization of these genes allowed us to determine their genomic structure within the sea cucumber genome (Figure 2). The structure of these in *H. glaberrima* is similar across the gene family, with all containing between 4 and 6 exons, with a minimum size of 59 bp (mostly first exons) and a maximum of 494 bp. All the annotated *Wnt* genes, with exception of *Wnt5*, had a first exon smaller than the rest, and the last exon was the largest of the studied *Wnts*. Different from the exon structures, intronic regions were variable across the family, showing a minimum of 636 bp and a maximum of 21,268 bp, with an average of 7363 bp. While to our knowledge, there is no information available about the structure of *Wnt* genes in other echinoderms, a few studies have described the structure of some of these in other species, such as the zebrafish *Brachydanio rerio* and humans [48,49]. *Wnt1* in *B. rerio* and humans shows a similar structure to that of *H. glaberrima*, i.e., composed of four exons. Similarly, the structures of human *Wnt5* and *Wnt16,* both with four exons, are the same as those found in *H. glaberrima.* Yet, when comparing the genomic structure of *Wnt2, H. glaberrima* contains six exons while humans have five exons.

We extended the characterization of the *Wnt* genes by performing a phylogenetic analysis of *Wnt* of various echinoderms species (Figure 3). This was done for two main reasons: (i) to confirm that each characterized *Wnt* was arranged in the same clade as its respective homologs, and (ii) to assess the conservation of these genes within the Echinodermata phylum. Our phylogenetic tree shows adequate grouping of *Wnt* with the expected clades; however, when looking deep into each of these, there were several interesting results. For instance, as one would expect, *Wnt* from holothuroids was shown to be in the same clade sister to that of echinoids. However, in the cases where partial sequences were utilized, the arrangement of *WntA* and *Wnt10* in *E. fraudatrix* was found to be different, resulting in an independent branch (*WntA*) or joining *P. miniata* as a sister clade to the echinoid clade (*Wnt10*). Other than this, which we attribute to the lack of a complete sequence, conservation was maintained, as all holothuroids and echinoid Wnt were maintained in distinct clades. Similarly, in almost all cases, *Wnt* genes of *P. miniata* appeared as a single branch throughout the tree.

Furthermore, there are variations between and within the assessed echinoderm classes regarding the loss or absence of individual *Wnt* genes (Figure 9). Seven subfamilies are conserved among the five classes: *Wnt3*, *Wnt5*, *Wnt7*, *Wnt9*, *Wnt10*, *Wnt16,* and *WntA*. In contrast, within the class Holothuroidea, eight subfamilies are conserved: *Wnt2*, *Wnt3*, *Wnt5*, *Wnt7*, *Wnt9 Wnt10*, *Wnt16,* and *WntA*. As can be seen in our phylogenetic tree, we did not include *Wnt10* and *Wnt16* of *A. japonicus,* as they were not deposited in NCBI. Nevertheless, we opted to include results from a previous publication [37] that investigated *A. japonicus Wnt* genes, albeit with some limitations. This study suggested that there is a duplication of *Wnt3*. However, both sequences deposited in NCBI (PIK62708.1; PIK45647.1) were 100% identical. Furthermore, the authors of that study suggested a duplication of *Wnt9*, but only one sequence could be found deposited. Additionally, we found four sequences identified as *Wnt8* in NCBI, two of which seemed to be duplications (PIK51024.1; PIK51023.1) and another with homology to *Wnt2* (PIK43830.1) in several species when performing a BLAST against the NCBI non-redundant database. Therefore, we only included the sequences for which we were certain of their homology, only one *Wnt3* and *Wnt9* sequences, *Wnt8* duplicates, and the *Wnt8* sequence that showed homology to *Wnt2* of other species (labeled here as *Wnt2*). Similarly, we used the *Wnt1* sequence from *A. japonicus* reported by Girich and colleagues (2019) [50], although this Wnt had not been previously reported in this species [37]. *Wnt8* was not found in *H. glaberrima.* Additionally, the *Wnt4* duplication appears to be unique to *H. glaberrima* compared to the species considered here. Lastly, the asteroid *P. miniata* is the only species that has all 13 Wnt subfamilies but no duplications.

Here, we also demonstrate the conservation of the *Wnt1, Wnt6, and Wnt10* cluster in *H. glaberrima.* This cluster is known to also contain *Wnt9* and *Wnt3* in several other species [47]. Nonetheless, the fragmented nature of our current draft genome limited the possibility of confirming the additional genes within the cluster. Still, we were able to manually confirm that in *H. glaberrima, Wnt1, Wnt6, and Wnt10* are clustered together (Figure 4). Our annotation showed that four exons of *Wnt1* were present in scaffold Hglab.02944 in the positive direction, the same scaffold that contains two exons of *Wnt6* in the negative direction, separated by 12.09 Kb. Similarly, we found that the scaffold Hglab.00936 contained the full *Wnt10* gene in the positive direction, along with the two first exons of *Wnt6* in the negative direction, separated by 39.46 Kb. However, this scaffold appeared to be in the opposite direction to scaffold Hglab.02944, for which, in Figure 4, we rotated it to demonstrate the clustering of *Wnt1, Wnt6, and Wnt10.* Moreover, we believe this to be the reason why the assembler utilized for the draft genome was not able to merge these two scaffolds. We also identified the same gene organization and direction of this cluster in the latest genome of *L. variegatus* (Figure 4), which are also clustered with *Wnt3* and *Wnt9* (data not shown). All of this is consistent with clusters previously reported in other species, such as *Drosophila melanogaster* [47]. However, the direction, and in some cases, the organization of the genes in such a cluster differs from species to species, while their grouping remains intact.

### 4.2. Wnt Signaling during Early-Stage Regeneration

Although correlative in nature, DGE analyses provide insight into which genes may participate in or modulate specific cellular processes. As for early-stage regeneration, which spans from 12-hpe to 7-dpe, *Wnt7* is downregulated at 12-hpe when injury and/or wound healing occurs. This contrasts with what was found in *A. japonicus*, where *Wnt7* was overexpressed during the injury/wound healing stage [37]. Upregulated genes during early-stage regeneration were *Wnt3*, *Wnt6* and *Wnt9*. These genes may be involved from 1- to 7-dpe when the mesothelium dedifferentiates and migrates to the mesentery, where the cells redifferentiate and proliferate to form the intestinal rudiment [27]. 

To determine which *Wnt*s may facilitate cellular processes during early-stage regeneration, we referenced previous work from our lab and others. First, it has been proposed that Wnt signaling is involved in proliferation [51], while a possible role of *Wnt*s in apoptosis and/or dedifferentiation has been discarded [33,51]. However, a specific Wnt has not been shown to be in control of cellular proliferation. Second, our data and other studies point to *Wnt6* as the most likely candidates to modulate the increase in proliferation since, in *H. glaberrima*, *E. fraudatrix*, and *A. japonicus, Wnt6* is up-regulated between 3- and 7-dpe. Lastly, this upregulation precedes the large spike in cell proliferation (~7-dpe) observed in the mesothelium of regenerating intestines [26,27], while the overexpression of *Wnt*3 and *Wnt*9 is stable throughout early-stage regeneration. Further evidence comes from other species. For example, *Wnt6* expression is most abundant in the vertebrate intestine in the crypt epithelium, where proliferation mostly occurs [20]. *Wnt6* is also a known contributor to tumorigenesis and the development of colon cancer via its effects on cell proliferation, apoptosis, and migration [52]. 

We cannot rule out *Wnt3* and *Wnt9* as possible controllers of proliferation during early-stage regeneration or during the injury/wound healing phase, particularly since the *Wnt9* protein is expressed in the regenerating mesothelium [31]. Additionally, *Wnt*s are known to have redundant functions [53]. Thus, *Wnt*3 and *Wnt*9 may supplement the effect of *Wnt*6. For example, *Wnt*3 is thought to time cell divisions in mammalian intestinal stem cells, where *Wnt6* and *Wnt9* are also expressed [54]. Alternatively, they could have their own independent function, possibly playing a role in the epithelial–mesenchymal transition (EMT) [27]. It is thus interesting that *Wnt3* can promote EMT in breast cancer cells [55] while in the sea cucumber, it is overexpressed beginning at 1-dpe, which is slightly earlier than the EMT timeline. Nonetheless, functional studies are required to be certain about the effects of these *Wnt*s during early-stage regeneration. 

### 4.3. Wnt Signaling during Late-Stage Regeneration

Late-stage regeneration is dominated by the formation of the lumen whereby at 14-dpe, luminal epithelial cells from the esophagus and cloaca proliferate and migrate into the intestinal rudiment [26]. At 21-dpe, proliferation continues, causing the intestine to elongate and grow in overall size. The *Wnt* genes most likely associated with these stages are *Wnt4b*, *Wnt5*, and *WntA*, all of which are uniquely overexpressed in late-stage but not in early-stage regeneration. The emergence of these *Wnt*s could be a result of the mesenchymal cells that surround the luminal epithelium. This is seen in mammals where mesenchymal cells beneath the basal lamina of the luminal epithelium uniquely secrete *Wnt4* and *Wnt5* [56]. However, the *Wnt*s associated with the formation of the rudiment (*Wnt3*, *Wnt6* and *Wnt9*) may also contribute to intestinal growth and elongation, since their upregulation persists during late-stage regeneration. 

It is also interesting that several *Fzd* genes are differentially expressed during late-stage regeneration, namely *Fzd1*, *Fzd4* and *Fzd10*. The high expression of *Fzd* genes is not fully unexpected since, in mammals, these receptors have been closely associated with the regeneration of the luminal epithelium and with stem cell modulation [57]. Additionally, another interesting finding from our work is the different patterns of gene expression between the anterior and posterior intestine. A similar finding had already been documented in *H. glaberrima*, i.e., differences in the spatial or temporal expression of genes and of cell proliferation and apoptotic events [58]. Additional examples of spatial differences in *Wnt* expression can be found in other tissues and other species, mainly during embryonic development. For example, in mouse embryos, *Fzd10* is found in the most posterior region of the epiblast and in the primitive streak but not in mesoderm migrating laterally and anteriorly [59]. In some spiders, *Wnt6* expression is not present posteriorly during embryonic patterning [60]. However, the most interesting of these is *Wnt5*, as it is expressed at the caudal end of the embryo during gastrulation and eventually in the distal portion of structures that extend from the primary body axis [61].

The overexpression of *WntA* during late-stage regeneration is also of particular interest. This *Wnt* is not found in vertebrates and is thought to have been lost during evolution. However, it is present in echinoderms, and its upregulation during late-stage intestinal regeneration has been confirmed in *A. japonicus* [34]. It is thus easy to speculate that *WntA* overexpression might be associated with the amazing regenerative abilities of echinoderms.

### 4.4. Wnt Signaling Pathways

We have also used our DGE analyses to elucidate the dynamics of Wnt signaling during holothurian intestinal regeneration. As mentioned, Wnt signaling can occur via the canonical Wnt/β-catenin pathway and the noncanonical Wnt/PCP pathway. Our data suggest that the cellular proliferation observed in early-stage regeneration occurs via the Wnt/β-catenin pathway, while luminal epithelial events in late-stage regeneration occur via the Wnt/PCP pathway. Our proposition is supported by comparative data, e.g., the *Wnt*s that are upregulated during early-stage regeneration have also been shown to be highly expressed in vertebrate paneth cells [17,20,62,63] and are known to activate the Wnt/β-catenin pathway [61,64,65]. Additional evidence comes from the downregulation of *Kremen* and *Groucho* in the early-stage regeneration. Kremen normally prevents Wnt from binding to Fzd, thus disabling the signaling cascade, while Groucho acts as a repressor of the LEF/TCF. Other evidence comes from our lab, where pharmacological experiments targeting the Wnt/β-catenin pathway both in vivo and in vitro modulate cell proliferation in early intestinal regenerate [51]. However, the most convincing evidence is the fact that RNAi for β-catenin decreases cellular proliferation in the early regenerating intestines in vitro [33].

As for late-stage regeneration, the Wnt/PCP pathway is most likely involved in the regeneration of the luminal epithelium. This is evidenced by a few observations: first, the upregulation of *Vangl*, which is known to facilitate gut elongation and lumen formation [66]; second, the upregulation of *Wnt5*, which can operate through the Wnt/PCP pathway to elongate the posterior gut, particularly when taking into account that, in mice, Wnt5 has been shown to work together with Vangl to orient cell division along the rostrocaudal axis to increase fore-stomach length [67]; third, *Fzd4*, which is associated with the Wnt/PCP pathway, was highly upregulated in late-stage regeneration, while it was under-expressed in early-stage regeneration; finally, the upregulation of *Kremen1*, which is an inhibitor of the Wnt/β-catenin pathway, suggests crosstalk between Wnt signaling pathways. This would also suggest that Wnt/PCP regulates Wnt/β-catenin during late-stage regeneration, so that it can impose its effects on cellular processes. 

We recognize that most DGE-based discussions are purely correlational, and further experimentation is necessary to conclude that specific genes have or lack a specified function. Nonetheless, our analyses can be viewed as a series of hypotheses that can be tested in future investigations.

In conclusion, the work presented here comprises an extensive analysis of sequence information ranging from the characterization of the *Wnt* genes found in the genome of our model organism, *H. glaberrima*, to a comparative expression analysis in normal and regenerating intestines. The results provide important insights into the molecular bases of intestinal regeneration and the role that Wnt signaling mechanisms might be playing in the process. They also provide investigators with a point of departure for molecular analyses, not only of regenerative organogenesis, but also of molecular evolution, signaling pathways, and organ homeostasis.

## Figures and Tables

**Figure 1 genes-14-00309-f001:**
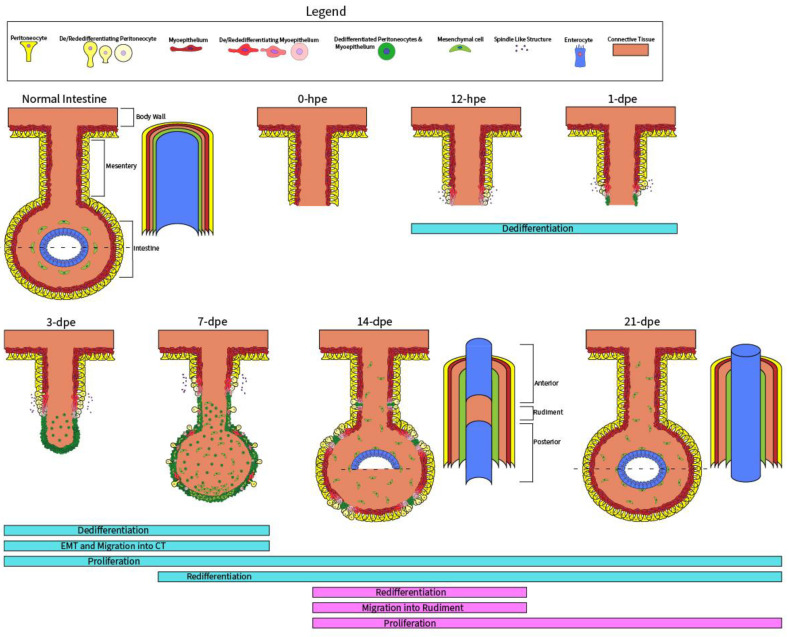
The process of intestinal organogenesis after evisceration. Beneath each timepoint is a color-coded cellular process unique to either the mesothelium (light blue) or luminal epithelium (dark pink). Normal Intestine: a normal intestine connected to the body wall with the mesentery and intestine distinguished, which are the controls for the rudiment and lumen, respectively. 0-hpe: the eviscerated intestine leaves a torn edge at the mesenterial tip, which is the site of injury response and initial wound healing, thus beginning the process of regeneration. 12-hpe: dedifferentiation occurs in the mesothelium, which is composed of peritoneocytes and myoepithelium; this process is characterized by the presence of spindle like structures that are discarded actin filaments from myoepithelial cells. 1-dpe: dedifferentiation of the mesothelium continues in a gradient along the mesentery, moving proximally toward the body wall. 3-dpe: some dedifferentiated cells ingress into the connective tissue (CT) of the rudiment, undergoing an epithelial-to-mesenchymal transition (EMT). 7-dpe: proliferation in the regenerating rudiment picks up pace; simultaneously, a few cells begin redifferentiating into mature mesothelium. 14-dpe: the rudiment is invaded by luminal epithelium derived from the esophagus (anterior) and the cloaca (posterior) that extend toward one another. The dashed line shows the anterior and posterior intestine with luminal epithelium and a surrounding mesenchyme (above dash), while in between is the rudiment without a lumen (below dash). 21-dpe: the anterior and posterior luminal epithelium connect to form a continuous lumen.

**Figure 2 genes-14-00309-f002:**
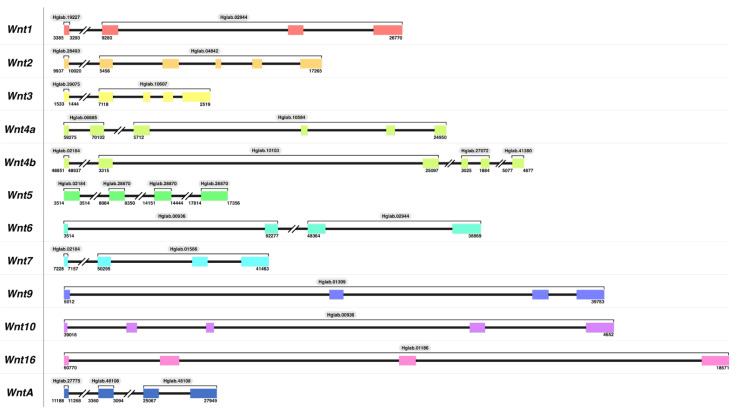
Structure of annotated *Wnt* family genes in *H. glaberrima.* The structure for all *Wnt* genes was characterized utilizing *Wnt* transcripts identified in the sea cucumber transcriptome data. All the scaffolds that contain each *Wnt* exon are identified by labels on top of each bracket (e.g., Hglab.02944). The start and end coordinates of the *Wnt*s span across each scaffold are shown at the bottom of each gene structure based on the genome nucleotide sequences (NCBI ID: GCA_009936505.2). Broken lines between exons represent a gene structure break due to genome fragmentation. Exon and intron sizes are based on base pair lengths, in which each base pair is equal to 0.2 for exons and 0.05 for introns.

**Figure 3 genes-14-00309-f003:**
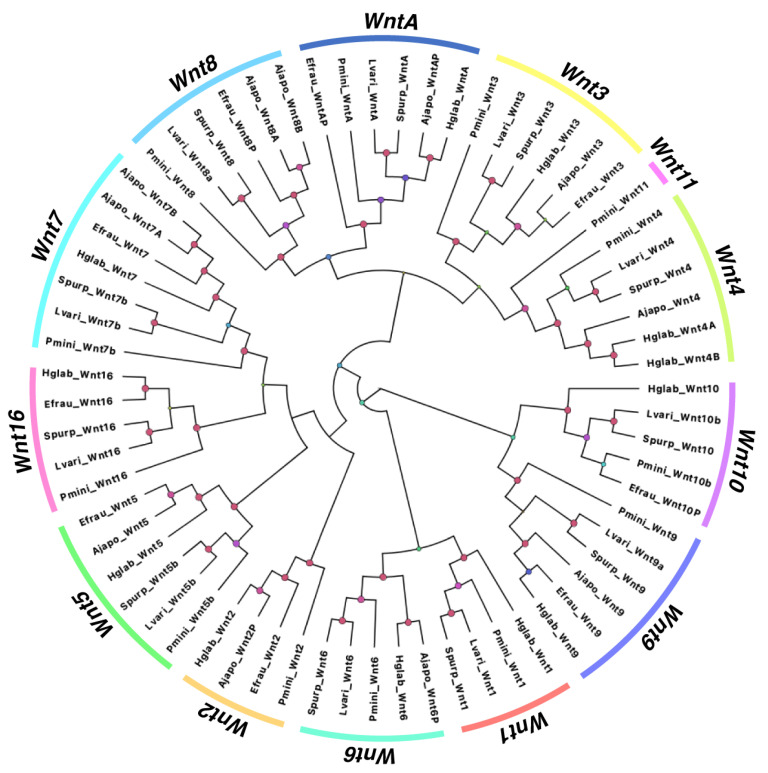
Phylogenetic analysis of *Wnt* from distinct Echinoderm species. Gene tree of all *Wnt* genes identified in *H. glaberrima* (Hglab), along with those deposited in NCBI or Echinobase for *S. purpuratus* (Spur), *L. variegatus* (Lvari), *P. miniata* (Pmini), *A. japonicus* (Ajapo), and *E. fraudatrix* (Efrau). For all partial sequences, a letter P was added to its label; all other letters are based on the gene name shown in NCBI. The size and color of circles in nodes are dependent on their bootstrap value. Nodes with bootstrap values over 95 are shown in pink.

**Figure 4 genes-14-00309-f004:**
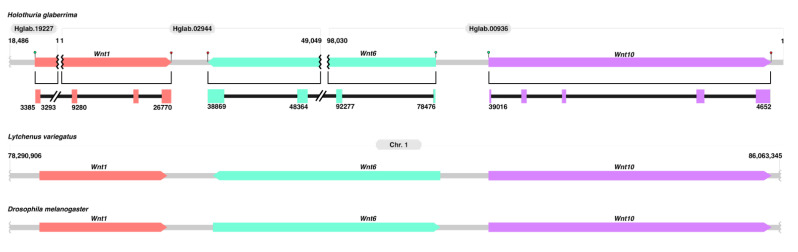
*Wnt* cluster conservation. Schematic illustrating the conservation of the *Wnt1-Wnt6-Wnt10* cluster in *H. glaberrima* compared to *L. variegatus* and *Drosophila melanogaster* (adapted from [47]). For *H. glaberrima*, we show the distribution of these genes across the draft assembly by complete squares with arrow heads pointing in the direction of the gene in the scaffold. The coordinates on top of the gene structures show scaffold size. The IDs on top of each gene structure represent the scaffold ID (NCBI ID: GCA_009936505.2). Separations of each *Wnt* due to fragmentation are shown as black line breaks. Start and stop codons are represented by green circles and red stars, respectively. The cluster of *L. variegatus* was characterized using its latest genome (Lvar_3.0; NCBI ID: 3495).

**Figure 5 genes-14-00309-f005:**
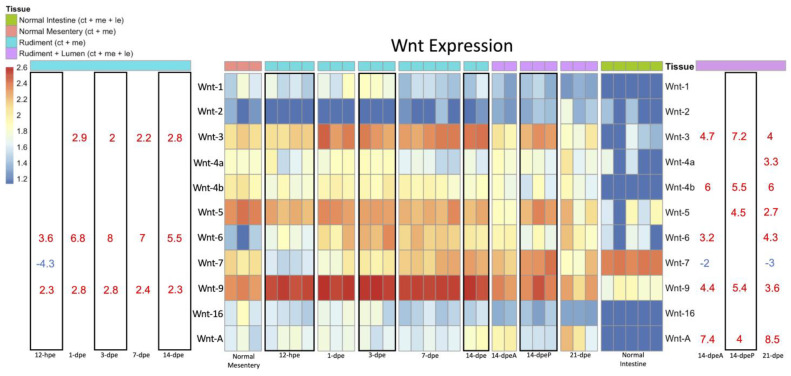
Heatmap of *Wnt* expression during early- and late-stage regeneration. This heatmap contains all the RNA-seq timepoints collected from our transcriptomic databases. The timepoints 12-hpe through 14-dpe should be referenced to Normal Mesentery, as this was the control. Similarly, the timepoints 14-dpeA, 14-dpeP, and 21-dpe should be referenced to Normal Intestine. On either side of the heatmap are the Log2foldchange (L2FC) values of a *Wnt* gene at a given timepoint. A significance threshold was set at L2FC <−2 or >2 with a pADJ value of 0.001. Above the columns are color coded labels that represent the tissue composition of the samples at a given timepoint. Abbreviations—CT: connective tissue, ME: mesothelium, and LE: luminal epithelium.

**Figure 6 genes-14-00309-f006:**
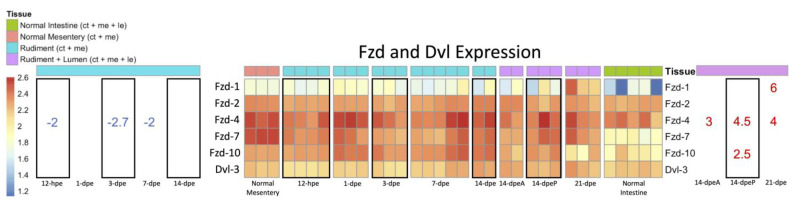
Heatmap of *Fzd* and *Dvl* expression during early- and late-stage regeneration. This figure can be read the same as the heatmap from Figure 5. The significance threshold is the same.

**Figure 7 genes-14-00309-f007:**
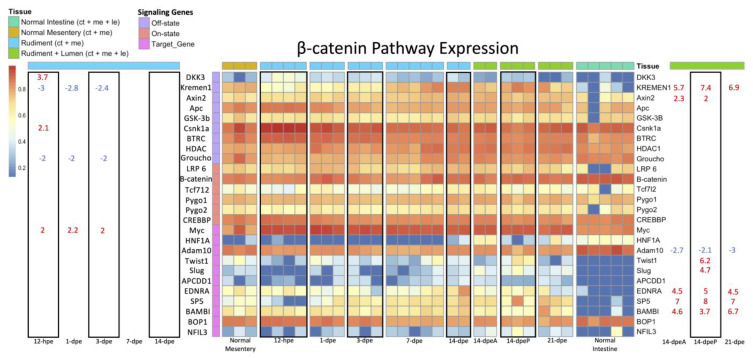
Heatmap of the signaling genes in the Wnt/β-catenin pathway. This figure is read the same as in Figure 5. The significance threshold remains the same. The only difference is that the rows are color coded to indicate the role of the gene in the Wnt/B-catenin pathway. Genes in the Off-state inhibit β-catenin signaling, while genes in the On-state facilitate Wnt/β-catenin signaling. The Target Genes are the downstream targets of the TCF/LEF transcription factor family that is turned on by β-catenin.

**Figure 8 genes-14-00309-f008:**
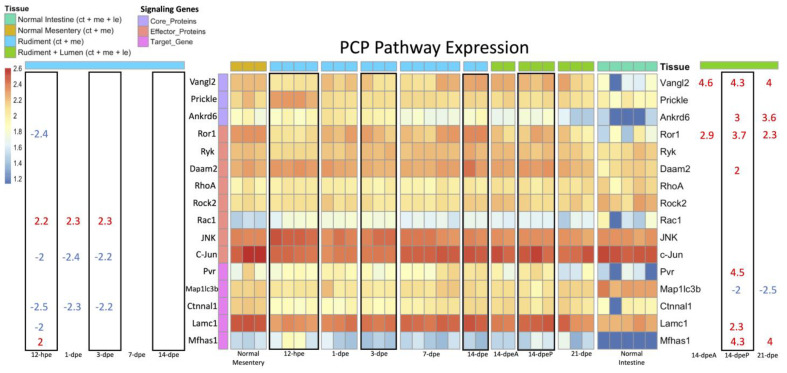
Heatmap of the signaling genes in the Wnt/PCP pathway. This figure is read the same as in Figure 5. The significance threshold remains the same. The only difference is that the rows are color coded to indicate the role of the gene in the Wnt/PCP pathway. Genes labeled as Core Proteins are unique and essential to the Wnt/PCP pathway and, therefore, are strong indicators of its presence. The effector proteins are genes involved in carrying out the signaling cascade, while target genes are activated by the transcriptions factor c-Jun.

**Figure 9 genes-14-00309-f009:**
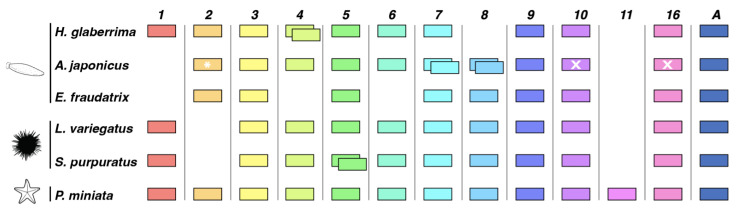
Summary of *Wnt* gene compositions of the echinoderm species assessed in this study. This schematic is based on a phylogenetic analysis of Figure 2. Asterisks depict manually characterized genes. White X depicts genes that could not be found in the NCBI database.

## Data Availability

All generate data will be available at NCBI during review. All the annotated Wnt genes have been deposited at NCBI (Accessions: BK062965, BK062966, BK062967, BK062968, BK062969, BK062970, BK062971, BK062972, BK062973, BK062974, BK062975, BK062976). All code utilized for the project is hosted at the following GitHub repository: https://github.com/devneurolab/HgWnt2023 (accessed on 4 January 2023). Draft genome is available at NCBI (ID: GCA_009936505.2).

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
