# Peer review of "Characterization and Expression of Holothurian Wnt Signaling Genes during Adult Intestinal Organogenesis"

_genes, 2023, doi:10.3390/genes14020309_

Round 1

Reviewer 1 Report

This manuscript looks well organized and scientifically sound. However, a few comments regarding the figures if it can be improved, because when it is enlarged it appears broken (increase the DPI or resolution).

In the method, the coordinates of the location on Google maps from taking sea cucumber samples must be given in detail and include the process of authentication and identification by a herbarium or the like!

Has the protocol for using animals in this study received ethical approval?

Reviewer 2 Report

In this manuscript, Auger et al characterized holothurian Wnt signaling genes and analyzed their expression during adult intestinal organogenesis using RNA-seq data. The manuscript is well written and highlighted the potential roles of holothurian Wnt family in adult organogenesis.  But the authors are requested to address the following concerns.

1. The introduction is not sufficient. Could the authors provide more information about Wnt signaling pathways? In this manuscript, three parts (Wnt genes, Wnt-associated genes, gens from Wnt/β-catenin and Wnt/PCP pathways) were involved. Please make their relationships clearly.

2. Please describe how to do RNA-seq library preparation and data analysis in section 2. Materials and Methods.

3. Could the authors provide NCBI accession number for RNA data and sequences used in this study?

4. The authors only compare Wnt signaling genes' expression based on RNA data.  The authors are strongly recommended to use qPCR to validate RNA data (several genes). 

Round 2

Reviewer 2 Report

Thanks for the authors' effors. The reviewer has no further concerns except for the following question. RNA-seq library preparation and data analysis are still not included in section 2. Materials and Methods.

2. Please describe how to do RNA-seq library preparation and data analysis in section 2.

Materials and Methods.

Response 2: The information on RNA-seq library preparation and analysis is now

included.

Author Response

Response to Reviewer 2 Comments (Round 2)

  1. Thanks for the authors' effors. The reviewer has no further concerns except for the following question. RNA-seq library preparation and data analysis are still not included in section 2. Materials and Methods.

                   Please describe how to do RNA-seq library preparation and data analysis in section 2. Materials and Methods.

Response 1: We have now expanded the protocol for RNA-seq and the differential expression analysis in the manuscript.